# Hypergraph Representation Learning from Noisy Node Attributes

1st Qian-Xi Tang
*the College of Computer and Data Science*
*Fuzhou University*
Fuzhou, China
qianxitang2000@163.com

2nd Chun-Yang Zhang
*the College of Computer and Data Science*
*Fuzhou University*
Fuzhou, China
zhangcy@fzu.edu.cn

*Abstract*—In recent years, hypergraph representation learning (HGRL) has become a focus of academic research, which aims to extract high-order topological patterns and attributes from hypergraph into low-dimensional representation vectors. However, most existing methods ignore the uncertainties in hypergraph data, thus failing to effectively leverage attribute features hidden in hypergraphs. For example, in a citation hypergraph, there exist uncertain semantics in node attributes regarding to papers. Therefore, we propose a new Fuzzy HGRL model, called FAHGN, which introduces fuzzy logic to grasp the attribute uncertainties. Specifically, the proposed FAHGN fuzzifies node attributes of the hypergraph as fuzzy input hypergraph signals, and makes full use of a spectral graph convolution operator to aggregate the fuzzy input signals to generate node representations. Through this way, it effectively considers feature-level uncertainties in hypergraphs and provides more expressive representations for effectively achieving downstream tasks. The experimental results on five real-world datasets validate the effectiveness of the proposed FAHGN against competitive baseline models.

*Index Terms*—Hypergraph, Fuzzy logic, Deep representation learning

## I. INTRODUCTION

Graph network have attracted increasing attention in recent years, leveraging pairwise relations defined between nodes to characterize the relationships between entities in real-world systems. However, the relationships among entities in reality extend far beyond simple pairwise connections, making it challenging to effectively model them with basic topological structures. For instance, in citation networks, a single paper may be cited by multiple other papers. Due to the inadequacy of traditional pairwise graph structures to accurately represent such higher-order group relationships, hypergraphs have emerged as a more powerful tool to depict these complexities. They provide more effective analytical insights for applications across various fields, including social network analysis [1], traffic flow control [2], product recommendation [3], knowledge graph completion [4], and biomolecular research [5].

To mine knowledge from hypergraph data, hypergraph representation learning serves as a promising solution, which aims to learn representation vectors for each node by extracting high-order topological patterns and attribute features hidden in hypergraphs. The concept of hypergraph learning, formulated as a propagation process on hypergraph structures, was first introduced by Zhou et al. [6], where spectral analysis methods on hypergraphs were employed. These methods typically involve rigorous mathematical derivations, which can confer a high degree of analytical precision. However, this rigor also imposes several limitations. In contrast, the advent of neural networks has reignited research into hypergraph learning. Feng et al. [7] were the first to propose hypergraph neural networks (HGNN), naturally extending the spectral methods of graph convolutional neural networks to hypergraphs and devising hypergraph convolution. Yadati et al. [8] proposes a new method of training graph convolutional networks on hypergraphs (HyperGCN), further incorporating hyperedge messages through intermediaries and presenting a generalized hypergraph Laplacian matrix. Johannes et al. [9] present a hypergraph contrastive learning framework (HyperCL), which leverages contrastive loss functions to capture both local and global structural information from hypergraphs. This method significantly enhances the robustness and generalization of hypergraph embeddings. Subsequently, numerous hypergraph-based methods have emerged and have been extensively applied in various domains such as computer vision [10], recommendation systems [11], and biochemistry [12]. These methods have achieved notable success and outperformed graph-based approaches, underscoring the potential value of in-depth research into HGRL.

However, existing hypergraph-based models overlook the uncertainties in the hypergraph data, particularly the uncertainties in attributes that are crucial for HGRL. For example, in citation hypergraphs, there exist uncertain semantics in node attributes regarding papers, which may interfere with the accuracy of the message-passing process. Additionally, typographical errors in citation networks may contaminate the node attributes of keywords, introducing noise into message aggregation and leading to indistinguishable representations. To address this issue, we propose a new HGRL model, called FAHGN, by incorporating fuzzy logic to address the uncertainties in attributes of hypergraphs. Specifically, the proposed FAHGN employs fuzzy logic systems to fuzzify the node attributes of hypergraphs as fuzzy input hypergraph signals, thereby capturing the vagueness of attribute semantics and alleviating the influence of noise in these attributes. Subsequently, spectral convolution operators are used for aggregating the fuzzy input hypergraph signals to generate node

representations. In this way, it sufficiently takes into account the attribute uncertainties in hypergraphs and provides more expressive representations for effectively achieving downstream tasks. Our contributions are summarized as follows:

- We propose a novel hypergraph representation learning model, referred to as FAHGN, which integrates fuzzy logic to effectively address the inherent uncertainties present in hypergraph data.
- This model utilizes fuzzy logic systems to facilitate the fuzzification of attributes, thereby capturing the inherent fuzziness of attribute semantics and mitigating the impact of noise associated with these attributes. Consequently, this enhancement leads to a more precise message passing and aggregation process within the hypergraph convolution framework, as implemented by a spectral hypergraph convolution network.
- Experimental results obtained from five real-world datasets validate the effectiveness of the proposed FAHGN, demonstrating its superiority compared to other strong methods.

The rest of the paper is organized as follow. Section II introduces the related work, in terms of hypergraph representation learning and fuzzy deep learning, followed by preliminaries presented in Section III. We illustrate the proposed FAHGN in Section IV, and report and analyze the experimental results in Section V. Finally, the conclusion of this work is described in Section VI.

## II. RELATIVE WORK

### A. Hypergraph learning

Hypergraphs demonstrate a stronger capability to model complex relationships among groups of nodes in real-world systems, with hyperedges capable of connecting an arbitrary number of nodes, when compared to traditional pairwise graphs. Recently, researchers have increasingly focused on hypergraph learning, achieving significant progress in various application scenarios, such as recommendation systems [11], biomolecular prediction in biochemistry [12], and sentiment analysis in natural language processing [13]. For the first time, Feng et al. [5] proposed a hypergraph neural network for learning data representations (HGNN), which incorporates a hyperedge convolution layer based on the hypergraph Laplacian, inspired by spectral graph neural networks. Yadati et al. [8] directly applied graph convolution operators [14] on hypergraphs in an effort to approximate hyperedges as pairwise edges. Dong et al. [15] suggested leveraging nonlinear activation functions to differentiate the significance of each hyperedge and node within a hypergraph, effectively controlling message passing. Addressing inductive problems related to unseen nodes, Arya et al. [16] proposed a novel message passing scheme to explore intra-hyperedge relationships and inter-hyperedge connections. Payne [17] also considered the joint influence of hyperedge context and permutation-invariant node attributes in inductive hypergraph representation learning. However, these methods frequently neglect the uncertainties inherent in hypergraphs, which may lead to suboptimal representations and, consequently, adversely impact downstream tasks.

### B. Fuzzy deep learning

Although various academic disciplines have benefited from deep learning models extracting feature representations from data in recent years, the high-dimensional complexity of data poses a primary challenge in data processing. Real-world data often contain noise and ambiguity, leading to fuzziness in the data [18], which presents significant difficulties for feature-based representations of data. Fuzzy logic, as a powerful tool, can assist deep learning models in compensating for shortcomings in handling fuzzy data [19]. Previous studies have demonstrated the effectiveness of combining fuzzy logic with deep learning in areas such as simplifying fuzzy rules [20], data processing [21], and handling data uncertainty [22]. Among these studies, the fuzzy neural network known as SVFNN [23], which utilizes support vectors as the basis for model classification, effectively combines fuzzy logic and neural networks to handle uncertain information, achieving high classification accuracy. Liu et al. [24] introduced a fuzzy peak neural network that integrates fuzzy logic with convolutional neural networks, effectively improving the performance of high-dimensional feature datasets. These studies demonstrate the potential of combining fuzzy logic with deep learning, opening up new avenues for processing real-world data and improving model interpretability. Therefore, in this work, we tend to combine fuzzy logic and HGRL to capturing the fuzziness of attribute semantics and weakening the influence of noisy attributes.

## III. PRELIMINARIES

### A. Hypergraph

Given a hypergraph denoted as $G(V, E, W)$, $V = \{v_1, v_2, ..., v_n\}$ denotes the set of n nodes, $E = \{e_1, e_2, ..., e_m\}$ represents a set of non-empty multi-element subsets of $V$ called hyperedges, and $W = \{w_{e,1}, w_{e,2}, .....w_{e,m}\}$ indicates the weights of $m$ hyperedges. For the node feature matrix of the hypergraph there is $X = x_1, x_2, ....x_n$, $X \in R^{n \times d}$, and for the hypergraph the hyperedge identity matrix has $E = e_1, e_2, ....e_m$, $E \in R^{m \times d}$. A hypergraph is usually represented by an incidence matrix, and the elements of the incidence matrix can be defined by

$$h(v, e) = \begin{cases} 1, & \text{if } v \in e \\ 0, & \text{if } v \notin e. \end{cases} \tag{1}$$

The node degree of a hypergraph indicates the weighted number of hyperedges connected to a node, and the node degree can be given by $d(v) = \sum_{e \in \mathcal{E}} w_e h(v, e)$. While the degree of a hyperedge represents the number of nodes connected to that hyperedge, which can be defined as $\delta(e) = \sum_{v \in \mathcal{V}} h(v, e)$. Notably, the node degree matrix $D_v$ and the hyperedge degree matrix $D_e$ are diagonal matrices consisting of node degrees $d(v)$ and hyperedge degrees $d(e)$, respectively.

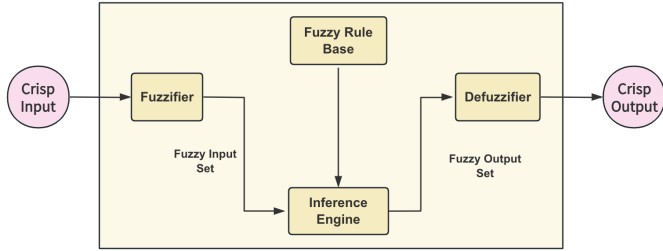

Fig. 1. The framework diagram of a fuzzy logic system.

### B. Fuzzy logic system

Fuzzy logic systems based on If-Then rules [25] are frequently employed in controlling algorithms and model developments, as they closely align with human perception and cognition. In this section, we will outline the general framework of fuzzy logic systems based on If-Then rules, as depicted in Fig. 1.

*Fuzzification* involves mapping the clear input values to a fuzzy set based on their membership function in a fuzzy logic system. A single instance fuzzifier is one of the simplest methods to compute the degree of membership, while more complex methods such as Gaussian, Trapezoidal and Trigonometric functions are more commonly used. Taking an example of Gaussian function, we give its definition as

$$F(\mu, \sigma, x) = \exp\left[-\frac{(x - \mu)^2}{2\sigma^2}\right], \qquad (2)$$

where $x$ is the input of the fuzzy system, while $\mu$ and $\sigma$ represent the mean and variance of the Gaussian function.

*Fuzzy Rule Base* is the core component of a fuzzy logic system that contains If-Then rules. After fuzzifying the inputs, we need to construct rules to combine and compute firing level using fuzzy logic operations for decision making. The rules of fuzzy logic contain a series of logical statements, usually consisting of If-Then parts. The If part is the premises (antecedents) and the Then part is the conclusion (consequent), which reveal the condition-consequence relationship. The rule structure is given by

$$R^i: \text{ If } x_1 \text{ is } F_1^i \text{ and } \cdots \text{ and } x_j \text{ is } F_j^i, \text{Then } y^i \text{ is } G^i, \quad (3)$$

where $x_j$ is the input of the fuzzy system, $y_i$ denotes the rule output, and $R^i$ represents the ith fuzzy rule. $G$ and $F$ are membership functions, $i = 1, ..., M, j = 1, ..., N$ where $M$ is the number of fuzzy rules and $N$ is the number of antecedents.

*Inference Engine* determines the degree of match between fuzzy inputs and rules, serving as the algebraic mechanism required to operate on fuzzy sets. Similar to traditional logical operations, basic fuzzy logic operations include union (s-norm), intersection (t-norm), and complement (c-norm). Defining two membership functions, $\mu_{F_1}$ and $\mu_{F_2}$ as

$$F_1 = \int_{x \in \mathbb{R}} \mu_{F_1}(x)dx, \quad F_2 = \int_{x \in \mathbb{R}} \mu_{F_2}(x)dx, \qquad (4)$$

their fuzzy logic operations are defined as

$$\mu_{F_1 \cup F_2}(x) = \max\left[\mu_{F_1}(x), \mu_{F_2}(x)\right], x \in \mathbb{R}$$
$$\mu_{F_1 \cap F_2}(x) = \min\left[\mu_{F_1}(x), \mu_{F_2}(x)\right]$$
$$\mu_{\bar{F}}(x) = 1 - \mu_F(x). \qquad (5)$$

*Defuzzification* aimes at transforming fuzzy conclusions into clear and specific outputs suitable for particular application scenarios. In the defuzzification process, the Centroid and the Center-of-Sets defuzzifiers are the most popular two techniques. The Centroid defuzzifier obtains clear values by identifying the centroid generated by the union of fuzzy sets of results. It samples $K$ points from result $G_{\text{out}}$, and calculates the centroid according to

$$y_{Centroid} = \frac{\sum_{i=1}^{N} y_i \mu_{G_{\text{out}}}(y_i)}{\sum_{i=1}^{N} \mu_{G_{\text{out}}}(y_i)}. \qquad (6)$$

As for the Center-of-Sets defuzzifier, it is necessary to determine the centroid of each fuzzy set of results, followed by computation to obtain clear values, which can be formulated as

$$y_{CoS} = \frac{\sum_{i=1}^{M} center^i g^i}{\sum_{i=1}^{M} g^i}, \qquad (7)$$

where $N$ represents the number of fuzzy rules, $g^i$ denotes the firing level of the $i-th$ rule, and $center^i$ signifies the center of the $i-th$ fuzzy set.

### IV. METHOD

In this section, we introduce the proposed FAHGN in detailed, including attributes fuzzification in hypergraph, spectral hypergraph convolution operator, and learning algorithm. The framework of the model is shown in Fig. 2.

### A. Hypergraph graph convolution with attribute fuzzification

Considering the limitations of existing methods as described in Section I, we introduce fuzzy logic to characterize the fuzziness of attributes and weaken the influence of attribute noises. Specifically, we first define a fuzzy membership function for each attribute elements of each node $x_i$ as

$$fuzzy_j(x_{i,j}) = \exp\left[-\frac{1}{2}\left(\frac{x_{i,j} - \nu_j}{\sigma_j}\right)^2\right], j = 1, 2, \ldots, d, \qquad (8)$$

where $\nu_j$ and $\sigma_j$ are the means and the standard deviations of the Gaussian membership functions, respectively. They are learnable parameters and can be initialized randomly. Notably, only one membership function is incapable to insufficiently the fuzziness of attributes. Therefore, we employ more than one membership function for each attribute, which can be defined as

$$F_{i,j,k}^l = fuzzy_j^k\left(x_{i,j}^l\right), k = 1, 2, \ldots, R, j = 1, 2, \cdots, d_l, \qquad (9)$$

to effectively explore the fuzziness of attributes and improve the variety of the message generation. After calculating the

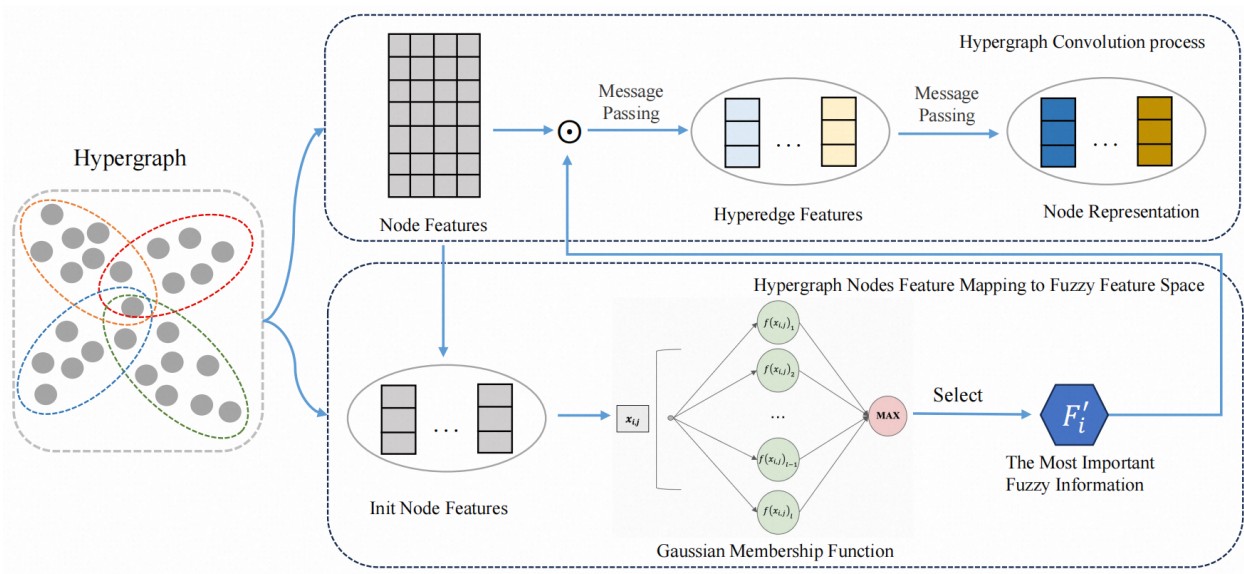

Fig. 2. For the initial hypergraph node features, we employ Gaussian membership functions to compute various degrees of membership. Each membership degree indicates the extent to which an element in the feature vector belongs to a particular fuzzy set. Subsequently, a maximum operator is employed to select the highest membership degree, representing the fuzzy set most likely corresponding to the feature vector. This step is regarded as extracting the most representative fuzzy information from different fuzzy sets. To integrate the original features with the fuzzy information, we apply the Hadamard product. The resultant fuzzy node features are progressively aggregated and abstracted during the hypergraph convolution process, ultimately forming the node representations.

degree of memberships of each attribute through multiple membership functions, the most important fuzzy information can be characterized through a $max$ operator, which can be defined as

$$F_{i,j}^{l}{}' = \max_{k=1,2,\ldots,R} \left\{ F_{i,j,k}^{l} \right\}, j = 1, 2, \cdots, d_l. \quad (10)$$

Then, we can extract the fuzzy message from attributes by merging the obtained fuzzy information with the attributes of the nodes, which can be given by

$$F_i^l = F_i^{l'} \odot x_i^l, \quad (11)$$

where $\odot$ represents the Hadamard merging operator. The the elements of obtained fuzzy messages have values between 0 and 1 due to the previous fuzzy operations, and they are effectively incorporated with fuzzy information. This fuzzy information will also be passed iteratively through the subsequent convolutional layers. In this work, we leverage a spectral hypergraph convolutional network to conduct message passing, and aggregate node representation for each node. The convolutional layer can be formulated as

$$F^{(l+1)} = \sigma \left( D_v^{-1/2} H W D_e^{-1} H^\top D_v^{-1/2} F^{(l)} \Theta^{(l)} \right), \quad (12)$$

where here $F$ is the fuzzy message matrix and $l$ denotes the index of layers.

### B. Learning Algorithm

Given the node representations generated by the proposed FAGHN, we use the cross-entropy loss function to measure the distance between the true label and the predicted label, which is given by

$$\mathcal{L}_{\text{sup-learning}} = -\frac{1}{N} \sum_{n=1}^{N} \sum_{k=1}^{K} y_{n,k} \log \hat{f}_{n,k} + \frac{\lambda}{2} \|w\|^2, \quad (13)$$

where $N$ is the number of training samples, $K$ denotes the number of classes of nodes, and $y_{n,k}$ represents the node's true label. $\hat{f}_{n,k}$ is the predicted label of the $n$-th sample, $\lambda$ control the model complexity and fuzziness level. Both terms in the formula are intended to ensure the performance of the model and the hypergraph convolution remain consistent.

TABLE I
DATASET STATISTICS

| | Cora | Citeseer | Pubmed | Cora-CA | DBLP |
|---|---|---|---|---|---|
| **Hypernodes** | 2,708 | 3,312 | 19,717 | 2,708 | 41,302 |
| **Hyperedges** | 1,579 | 1,079 | 7,963 | 1,072 | 22,363 |
| **Features** | 1,433 | 3,703 | 500 | 1,433 | 1425 |
| **Classes** | 7 | 6 | 3 | 7 | 4 |
| **Max $d_e$** | 5 | 26 | 171 | 43 | 426 |
| **Min $d_e$** | 2 | 2 | 2 | 2 | 3 |
| **Max $d_v$** | 145 | 88 | 281 | 45 | 373 |
| **Min $d_v$** | 1 | 1 | 1 | 1 | 1 |

## V. EXPERIMENTS

To verify the effectiveness of the proposed FAHGN, we conduct experiments on five real-world datasets in the node classification task.

| Method | Co-Cora | Co-Citeseer | Pubmed | Cora-CA | DBLP-CA |
|--------|---------|-------------|--------|---------|---------|
| GCN* | $77.11 \pm 1.8$ | $66.07 \pm 2.4$ | $82.63 \pm 0.6$ | $73.66 \pm 1.3$ | $87.58 \pm 0.2$ |
| GAT* | $77.75 \pm 2.1$ | $67.62 \pm 2.5$ | $81.96 \pm 0.7$ | $74.52 \pm 1.3$ | $88.59 \pm 0.1$ |
| HGNN | $79.39 \pm 1.8$ | $72.45 \pm 2.3$ | $86.44 \pm 0.7$ | $83.24 \pm 1.2$ | $91.03 \pm 0.3$ |
| HyperConv | $76.19 \pm 2.1$ | $64.12 \pm 2.6$ | $83.42 \pm 0.6$ | $73.52 \pm 1.0$ | $88.83 \pm 0.2$ |
| HCHA | $79.14 \pm 1.0$ | $72.42 \pm 2.1$ | $86.41 \pm 0.4$ | $83.65 \pm 1.0$ | $90.92 \pm 2.2$ |
| HNHN | $76.21 \pm 1.7$ | $72.64 \pm 2.2$ | $80.97 \pm 0.9$ | $77.19 \pm 1.6$ | $86.71 \pm 1.2$ |
| HyperGCN | $78.45 \pm 7.4$ | $59.92 \pm 9.6$ | $78.40 \pm 9.2$ | $60.65 \pm 9.2$ | $76.59 \pm 7.6$ |
| HyperSAGE | $64.98 \pm 5.3$ | $52.43 \pm 9.4$ | $79.49 \pm 8.7$ | $64.59 \pm 4.3$ | $79.63 \pm 8.6$ |
| UniGCN | $78.81 \pm 1.9$ | $73.05 \pm 1.9$ | $88.25 \pm 0.7$ | $83.60 \pm 1.4$ | $91.31 \pm 0.2$ |
| AllSet | $78.59 \pm 1.7$ | $73.08 \pm 1.8$ | $\mathbf{88.72 \pm 0.9}$ | $83.63 \pm 1.2$ | $91.53 \pm 0.3$ |
| **FAHGN** | $\mathbf{80.21 \pm 1.2}$ | $\mathbf{73.35 \pm 1.1}$ | $86.99 \pm 0.6$ | $\mathbf{84.37 \pm 1.1}$ | $\mathbf{91.79 \pm 0.2}$ |

## A. Datasets

*Cora*, *CiteSeer* and *Pubmed* are the co-citation datasets derived from Cora, Citeseer and Pubmed databases [26].

The co-citation dataset preprocesses the papers in the data topics to form a binary bag-of-words vector as the feature description information of each node.

*Cora-CA* and *DBLP* datasets are the co-authored datasets collected from Cora database and DBLP database, respectively [27]. The nodes in the dataset represent the hyperedge indicate the relationship among all the papers related to an author. The DBLP dataset comes from six different conferences, including "Algorithms", "Databases", "Programming", "Data Mining", "Intelligence", and "Vision", and the abstracts of all the papers in the dataset are extracted in order to compose the node attributes of the DBLP dataset. Specifically, given abstracts of all the papers, a dictionary of the most frequently used words, with a frequency of more than 100, is constructed. Based on this dictionary, a bag-of-word vector with 1425 dimensions is generated for each node as their node attributes. The statistics of these four datasets are shown in TABLE I.

## B. Baseline Model

A number of recent works have demonstrated the strong competitiveness of hypergraph representation learning [28]. Therefore, we compare the models with a variety of strong baseline models including the two classical graph-based models, GCN [14] and GAT [29], and eight state-of-the-art HGNN [7], Hyperconv [30], HCHA, HNHN [15], HyperGCN [31], HyperSage [16], UniGCN [32], ALLSet [28]. Notably, GCN and GAT are unable to process hypergraphs directly, and thus we use the cluster expansion method [33] to expand hypergraphs into graphs before using them.

## C. Node Classification

Node classification task is one of the most popular tasks in graph network analysis, in which the model needs to predict which category each node belongs to, here *Accuracy* is used as the evaluation metric for classification task, which represents the ratio of the number of correctly classified samples to the total number of samples. Note that, the higher the accuracy, the better the model classification ability. The data used for training, validation and test are set by 50%, 50% and 25% randomly selected nodes, respectively. To obtain credible and stable results, we run 50 times indepent experiments with random splits training, validation and test data. The results of the node classification are presented in TABLE II.

The proposed FAHGN outperforms almost all baseline models in all four datasets. This is mainly attributed to the fact that FAHGN fully considers the influence of fuzziness and noise in attributes in the message propagation process, removing the uncertainties in hypergraphs. By continuously correcting the model representation during message propagation, the proposed FAHGN improves the quality of node representation, thus improving the overall performance. Although the performance on the *Pubmed* dataset is not excellent enough, this result is understandable since each model has its specific strengths and weaknesses and cannot outperform other models in all cases. The results of the node classification are presented in TABLE II.

## VI. CONCLUSION

We have proposed a novel hypergraph representation learning model, called FAHGN, that incorporates fuzzy logic into hypergraph representation learning to address the uncertainties of hypergraph attributes. This model, inspired by fuzzy logic systems, has fuzzified the attributes of hypergraphs to generate fuzzy messages and has employed spectral hypergraph convolution networks to aggregate the fuzzy messages layer by layer to obtain the node representations. This approach has effectively modeled the fuzziness of attributes and has significantly reduced the uncertainties in attributes, thus improving the quality of the learned representations of nodes in hypergraph representation learning. Experimental results have demonstrated that the proposed FAHGN outperforms existing models on several benchmark datasets, showcasing its strong competitiveness and validating the effectiveness and superiority of our motivation.

Despite its outstanding performance, the proposed FAHGN has certain limitations. For instance, when dealing with large-

scale hypergraphs, computational complexity and resource demands have become its bottlenecks. Additionally, the model has exhibited high sensitivity to parameter selection, with optimal parameter settings varying significantly across different datasets. This may have further aggravated computation complexity. These limitations have induced two possible avenues for future work. First, exploring more efficient computational methods to handle large-scale hypergraph data; second, investigating automated hyperparameter tuning techniques to reduce dependency on parameter settings.

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
