# OpenReview forum: "Hypergraph Representation Learning from Noisy Node Attributes"
_IEEE.org/ICIST/2024/Conference — IEEE ICIST 2024 Conference Submission_

### Official Review · Reviewer_5uWh · 2024-08-21
**Interesting work**

**Rating:** 7
**Confidence:** 4

**Review:**

The paper titled "Hypergraph Representation Learning from Noisy Node Attributes" proposes a new Fuzzy HGRL model to introduce fuzzy logic to grasp the attribute uncertainties. And the experimental results on five real data sets validate the effectiveness of the proposed FAHGN for the competitive baseline model. My specific feedback is as follows: 1) Some formatting issues need to be addressed. 2) The future work is missing in the section of Conclusion.

---

### Official Review · Reviewer_swJZ · 2024-08-22
**This article is very interesting and a good one**

**Rating:** 7
**Confidence:** 3

**Review:**

This study proposed a new hypergraph representation learning model, named FAHGN, by introducing fuzzy logic to address the uncertainties in hypergraph data. The theory is correct and can be accepted after responding the following comments.
(1) In the introduction, it is not enough to state the current work. It should be expended and reconstructed.
(2) There are many typos and grammar errors. The authors should have a native English speaker or software packages to perform the editing check.
(3) The font size of TABLE II needs to be modified to make the article more aesthetically pleasing
(4) The conclusion of the article suggests using the existing completion tense for description.

---

### Official Review · Reviewer_XYuh · 2024-08-28
**This paper proposes a new fuzzy hypergraph representation learning model, which introduces fuzzy logic to grasp the attribute uncertainties. Specifically, the proposed FAHGN fuzzifies node attributes of the hypergraph as fuzzy input hypergraph signals, and makes full use of a   pectral graph convolution operator to aggregate the fuzzy input signals to generate node representations. In my opinion, there are some comments that should be considered during the revision of the manuscript which are listed below.**

**Rating:** 7
**Confidence:** 3

**Review:**

1. What are the significant differences between this study and previous studies? The author needs more explicit emphasis.
2. The conclusion needs to use the present perfect tense to fit the logic of the article.
3. Is it reasonable that 'Deep representation' appears only in key words in the text? The author needs to give a reasonable explanation.

---

### Comment · Reviewer_swJZ · 2024-08-21
**This article is very interesting and a good one**

This study proposed a new hypergraph representation learning model, named FAHGN, by introducing fuzzy logic to address the uncertainties in hypergraph data. The theory is correct and can be accepted after responding the following comments. (1) In the introduction, it is not enough to state the current work. It should be expended and reconstructed. (2) There are many typos and grammar errors. The authors should have a native English speaker or software packages to perform the editing check. (3) The font size of TABLE II needs to be modified to make the article more aesthetically pleasing (4) The conclusion of the article suggests using the existing completion tense for description.

---

### Decision · Program_Chairs · 2024-09-06

Accept (Oral)